# A Novel 3D-Morphology Pyrene-Derived Conjugated Fluorescence Polymer for Picric Acid Detection

**DOI:** 10.3390/nano12224034

**Published:** 2022-11-17

**Authors:** Yu Fan, Yang Chen, Yueling Bai, Baoli An, Jiaqiang Xu

**Affiliations:** NEST Lab., Department of Chemistry, Shanghai University, Shanghai 200444, China

**Keywords:** conjugated polymer, fluorescence titration, picric acid detection, restriction of intramolecular rotations (RIR), DFT calculation

## Abstract

Aggregation-induced quenching (ACQ) is a hard problem in fluorescence material, leading to a poor utilization rate of fluorophores. In this work, 1,3,6,8-tetrakis(4-formylphenyl)pyrene (TFPPy) was synthesized and used as a precursor to build two kinds of fluorescence polymer. The TFFPy molecule with D2h symmetry can easily form polymers with C3 symmetry amines through the Schiff base reaction, making the resulting polymer a 3D amorphous material. Thus, ACQ of fluorophore can be reduced to minimum, making the most usage of the fluorescence of pyrene core. Fluorescence titration and DFT calculation can clearly prove this conclusion. The resulting CPs showed a highly sensitivity to picric acid, down to 3.43 ppm in solution, implying its potential in explosive detection.

## 1. Introduction

The detection of nitroaromatic explosives has been a topical issue for decades [1,2]. Among various nitroaromatic explosives, picric acid (PA), a more powerful explosive than 2,4,6-trinitrotoluene(TNT) [3], is widely used in leather, dye, pesticides and plastic dissolution. However, PA is highly toxic and dangerous for human beings and the environment when it is released [4]. Therefore, more and more attention is focused on detecting PA with a highly efficient, low-cost, specific, sensitive and fast response. Currently, detection methods for nitroaromatic explosives known widely include fluorescent probes [5,6,7], gas chromatography [8], ion mobility spectrometry [9] and electrochemical detection [10,11].

Among these various methods, fluorescence-based sensing methods have the advantage of being highly sensitive, convenient and cost-effective, and widely focused on by academy and industry. A variety of fluorescent materials are adopted for fluorescent chemosensors, for example, conjugated polymers [12,13], luminescent metal-organic frameworks [14,15,16], and nanomaterials [16,17]. Among them, aromatic conjugated polymers (CPs) are a very important kind of organic polymers used as fluorescent chemosensors due to the advantages of signal amplification, ease of device fabrication and the ability to combine different outputs. Although there has been fast progress in the development of fluorescent CPs, there remain challenges in sensing materials with high efficiency, at low cost, and in an eco-friendly way. Moreover, the design strategy of building CPs should also be developed.

During the development of fluorescence materials, aggregation-caused quenching (ACQ) phenomenon was an important discovery in 1954 by Förster [18]. The photophysical processes and working mechanism of fluorophores in fluorescence materials have been deeply revealed during half a century’s research [19,20,21]. When fluorophore and nearby aromatic rings aggregate in paralleling, the excited state of such aggregates decays or relaxes back to the ground state via non-radiative channels, resulting in the emission quenching of the fluorophores; this is called aggregation-cased quenching (ACQ) and is a very common phenomenon.

Nowadays there is a famous kind of conjugated polymer, covalent organic frameworks (COFs), the development of which has flourished during the past 15 years [22,23]. Through strict topological chemistry, very beautiful structures have been synthesized and made use of in many fields. In 2D-COFs, a variety of fluorophores are synthesized and employed as building blocks, such as tetraphenyl ethylene, pyrene, porphyrin, and hexahydroxytriphenylene (HHTP). These building blocks are adopted due to their planar characteristic and topological symmetry, which are the fundamental features of 2D-COFs. In consideration of ACQ phenomenon, however, 2D COFs may cause fluorescence quenching by aggregation or π-π stacking, thus restraining the luminescence of fluorophores. It is therefore very important to find a way to eliminate the negative impact of the ACQ phenomenon. If fluorophores can be used to build 3D structures in which all the aromatic fluorophore groups are not parallel with each other, the ACQ effect will be reduced.

## 2. Material Preparation

In this study 1,3,6,8-tetrakis(4-formylphenyl)pyrene (TFPPy) was synthesized and used as a fluorophore to build CPs with amine molecules through the Schiff base reaction. The monomer of TFPPy was obtained through a Suzuki coupling according to the literature [24] with a modified solvent condition (Figure 1). Two kinds of CPs were grown with a solvothermal method by using TFPPy as an aldehyde monomer. The amine monomer is melamine for CP-shu-1, and 4,4′,4″-(1,3,5-triazine-2,4,6-triyl)trianiline (TTTA) for CP-shu-2 (Figure 2). Since the D2h symmetry of TFPPy and the C3 symmetry of two kinds of amine molecules cannot form a 2D structure in topological chemistry, only a 3D structure can form. To our knowledge, the Schiff base reaction is equalized, and the equilibrium reaction rate will rise in the presence of a weak acid such as acetic acid. We attempted to implement the Schiff base reaction in without the presence of acetic acid, so that the reverse reaction rate decreases and the final structure can be irregular.

1,3,6,8-tetrabromopyrene (TBP): Synthesized through bromination of pyrene according to the literature [24].

1,3,6,8-tetrakis(4-formylphenyl)pyrene (TFPPy): Synthesized through a Suzuki coupling according to the literature [24] with a modified solvent condition.

A mixture of TBP (2.59 g, 5.0 mmol), 4-formylphenylboronic acid (3.40 g, 22.7 mmol), Tetrakis(triphenylphosphine)palladium (0) (0.49 g, 0.4 mmol), and K_2_CO_3_ (3.45 g, 25.0 mmol) in THF (200 mL) and H_2_O (25 mL) was stirred under argon atmosphere for 3 days at 100 °C. After cooling to room temperature, the yellow suspension reaction mixture was poured into a solution of ice containing concentrated hydrochloric acid. The yellow solid was filtered and washed with 2 M HCl (50 mL) three times. The product was extracted with CHCl_3_ (3 × 100 mL) and dried over MgSO_4_. After filtration, the solvent was removed under reduced pressure and the resultant solid residue was recrystallized from hot CHCl_3_ to yield TFPPy as a bright yellow powder (2.28 g, 74%).

Preparation of CMP-shu-1: TFPPy (185.0 mg, 0.30 mmol), melamine (50.4 mg, 0.40 mmol) and o-dichlorobenzene (5 mL) were added into a 10 mL glass seal tube. The resulting suspension was sonicated for 5 min and then degassed through three freeze-pump-thaw cycles. After that the tube was sealed under an argon atmosphere and then warmed to room temperature. The sealed tube was kept at 150 °C without disturbance for 3 days. After cooling down to room temperature, the suspension was filtered and the residue was washed with a large amount of dichloromethane and acetone and then dried under dynamic vacuum at 120 °C for 2 h to yield a pale yellow powder (202.9 mg, 87.0%).

Preparation of CMP-shu-2: TFPPy (185.0 mg, 0.30 mmol), 4,4′,4″-(1,3,5-triazine-2,4,6-triyl)trianiline (TTTA) (141.6 mg, 0.40 mmol) and 1,4-dioxane (5 mL) were added into a 10 mL glass seal tube. The resulting suspension was sonicated for 5 min and then degassed through three freeze-pump-thaw cycles. After that the tube was sealed under an argon atmosphere and then warmed to room temperature. The sealed tube was kept at 120 °C without disturbance for 3 days. After cooling down to room temperature, the suspension was filtered and the residue was washed with a large amount of dichloromethane and acetone and then dried under dynamic vacuum at 120 °C for 2 h to yield a pale yellow powder (274.2 mg, 84.6%).

## 3. Characterization and Sensing Experiment

The formation of CPs from the condensation reactions of the monomers was supported by FT-IR (Figure 3) and ^13^C CP-MAS NMR spectra (Figure 4). FT-IR spectra reveal the disappearance of peaks corresponding to NH_2_ around 3400 nm^−1^, and the formation of peaks around 1620 nm^−1^ corresponding to the C=N double bond, which can prove the success of the Schiff base reaction, which is the evidence of the condensation reaction. The formation of CPs was also proved by ^13^C CP-MAS NMR spectra, where the C=N signal is located at 187 ppm. Thermogravimetric analysis (TGA) shows that 5% weight loss occurred at 378 and 393 °C for CP-shu-1 and CP-shu-2, respectively (Figure 5), indicating that they have high thermal stabilities. Powder X-ray diffraction (PXRD) profiles of the CPs do not exhibit obvious peaks, demonstrating that the structures of CPs materials are amorphous (Figure 6).

To observe the 3D structure of as-prepared CPs materials, scanning electron microscopy (SEM) and transmission electron microscopy (TEM) were performed. As shown in Figure 7, the SEM of both CP-shu samples reveal a morphology of nanorods. Since the combination of C_2_V symmetry for aldehyde monomer and C_3_ symmetry for amine monomer cannot result into 2D planar structure, only a 3D nanorod structure can be formed. It should be noticed that the average rod diameter of CP-shu-1 is about 100 nm, far less than CP-shu-2, which is about 1 μm. This might because that the melamine has a short length of “arm” while forming polymer, resulting in a narrow and crowded microstructure. The rod diameters of two CPs were also observed by TEM, indicating that the average thickness of CP-shu-1 is about 200 nm, which is smaller than that of CP-shu-2, which is about 500 nm. TEM also demonstrated that the nanorods are amorphous and irregular, which is probably because of the irreversible reaction condition of the Schiff base reaction.

To evaluate the fluorescence properties, solid-phase fluorescence spectra of CPs were performed using the as-prepared powder (Figure 8). After excitation spectra were tested, 336 nm and 499 nm were chosen as excitation wavelengths for CP-shu-1 and CP-shu-2, respectively. As shown in Figure 8, the emission density for CP-shu-1 shows 0 before 499 nm, so there is almost no influence on excitation wavelength choice. The emission peaks for CP-shu-1 and CP-shu-2 are 540 nm and 556 nm, respectively. There is a 16 nm red shift from CP-shu-1 to CP-shu-2, mainly resulting from the triazine ring in melamine being an electron-withdrawing group, which may enlarge the band gap of adjacent pyrene, causing a blue shift. While in TTTA, the three benzene rings connected with the centered triazine ring can reduce this effect by a π-π conjugated bond. The emission intensity of CP-shu-1 is 1.5 times greater than that of CP-shu-2, indicating the CP-shu-1 exhibits better fluorescence. This might be because in CP-shu-2 there is one more benzene ring that can do relatively free rotation in its polymer chain. According to the aggregation-induced emission (AIE) theory [25], the restriction of intramolecular rotations (RIR) can enhance emission. As the extension, therefore, the increasing amount of freely rotating benzene rings can reduce emissions.

The fluorescent properties of CP-shu-1 and CP-shu-2 were further investigated by fluorescence titration. Before titration, the CPs were dispersed in THF via ultrasound for 10 min to make a 0.2 mg/mL suspension. As shown in Figure 9, different changes came out while different kinds of BTEX reagent were added into the suspension. With the addition of benzene, the fluorescence emission intensity is enhanced. With the addition of nitrobenzene, in contrast, the fluorescence intensity decreases to almost zero. For benzene, AIE theory can also be used to explain emission enhancement. For the pyrene-based CP frameworks, there are four spin-free benzene rings connecting with the pyrene core which may still consume the exciton energy and increase the non-radiative decay rate. There remains, therefore, the potential capability to make a better emission, which can be proved by the fluorescence absolute quantum yield (QY) measurement, which showed 14.69% and 9.37% for CP-shu-1 and CP-shu-2, respectively. Such a low QY is induced by the rotation of relatively free benzene rings near the pyrene. Furthermore, there are more free benzene rings in CP-shu-2 than in CP-shu-1, with the result that the QY for the CP-shu-2 is lower than for CP-shu-1. When adding benzene or toluene into the suspension, the benzene or toluene molecules can easily stack with the benzene rings near pyrene, resulting in an aggregation-induced emission(AIE) by RIR, thus further making a larger fluorescence intensity. In contrast, for nitrobenzene, the intermolecular charge transfer (ICT) mechanism can sufficiently explain the quenching phenomena. While the CP structures contain a rich content of N element, they can easily exchange electrons with electron-deficient Lewis acid, resulting in a fluorescence quenching.

Since the fluorescence of CPs can be quenched by electron-deficient nitro-compound, we further explore the application of CPs as a fluorescent chemosensor in picric acid (PA) detection (Figure 10). Fluorescence spectra of the CPs in the presence of picric acid were recorded. Before recording, the CPs were dispersed into acetonitrile to obtain a suspension liquid at a concentration of 0.2 mg/mmol. As can be seen in Figure 10, with the addition of PA, there was a rapid decrease in the fluorescence intensity of CPs. Significant fluorescence quenching of 12.3% and 9.7% were observed even in the presence of 5 ppm PA for CP-shu-1 and CP-shu-2, respectively. According to the plotted equation (I_max_ − I)/(I_max_ − I_min_) vs. lg[PA] [26], the limit of PA detection was calculated to be 3.43 ppm and 3.76 ppm for CP-shu-1 and CP-shu-2, respectively.

## 4. Mechanism Analysis

To prove the fluorescence mechanism of CPs, DFT calculations (Figure 11) were cast with the Gaussian 09 package to optimize structure and calculate the HOMO-LUMO orbitals of CPs at the B3LYP/6-31G level without consideration of solvation. The approximate alternative model is used to substitute the large complicated polymer, containing an aldehyde pyrene molecule and an amine molecule coupled with the C=N double bond. As shown in Figure 11, both of the CPs showed the HOMO orbital at the pyrene core, and the LUMO orbital shifted to the C=N linkage arm. When the CPs are excited by photons, the excited photoelectron jumps into the LUMO orbital, which is partially located on free-spinning benzene rings (Figure 11b,d). The energy of the photoelectron is partially decreased through the free rotation, resulting in the low QY. Moreover, this can also explain the mechanism of the benzene titration experiment in Figure 9. Acetonitrile solvent cannot prevent the RIR effect because of the low size and long shape of the acetonitrile molecule. While adding benzene into the suspension, the benzene can perfectly fit the free-spinning benzene arm in which the LUMO is located, causing the π-π stacking effect and increasing the difficulty of rotating, thus enhancing the fluorescence of CPs.

The spin energy of TFFPy was further calculated to investigate in depth the rotating of the benzene arm (Figure 12). Because 1 Ha equals 2625.48 eV, the spinning of the benzene ring is calculated to be 0.015619 Ha, or 0.425eV. This can prove that the RIR effect happens with the emission and excitation process of CPs in Figure 8. For CP-shu-1, the average excitation wavelength is 450 nm, the emission wavelength is 540 nm, and the energy difference between emission and excitation is 0.459 eV. For CP-shu-2, the spinning energy is lowered to only 0.254 eV, entirely because there is a more easily spinning part in TTTA that links with the TFFPy. This calculation, therefore, can prove that there is precisely an RIR effect in CPs.

## 5. Conclusions

In conclusion, two kinds of new pyrene-derived CP materials have been designed and synthesized for the fluorescence detection of picric acid. The 3D structure of CPs can avoid the ACQ phenomenon, thus making the greatest usage of the pyrene fluorescence. The limit of detection toward PA is calculated to be 3.43 ppm for CP-shu-1 and 3.76 ppm for CP-shu-2, respectively, indicating their good potential in PA detection. Moreover, since most fluorophores are planar, we believe this 3D-structure strategy that can effectively avoid ACQ phenomena will provide new guidance for the design of new fluorescence materials in the future.

## Figures and Tables

**Figure 1 nanomaterials-12-04034-f001:**
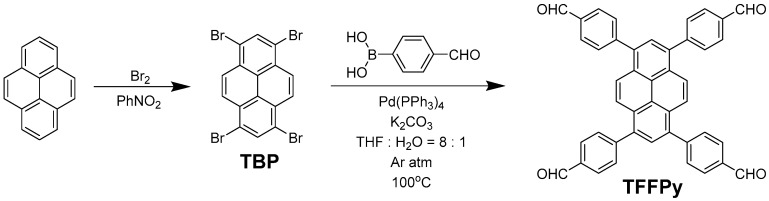
The preparation of TFFPy.

**Figure 2 nanomaterials-12-04034-f002:**
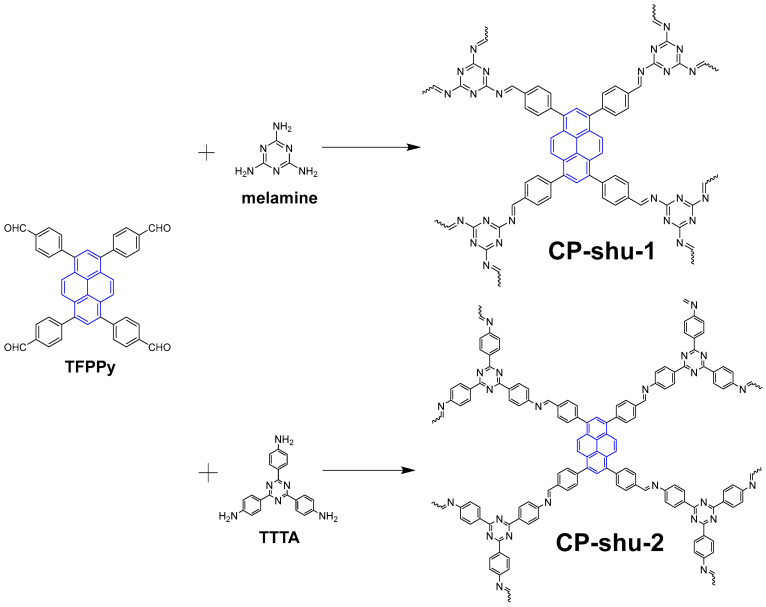
The preparation of CP-shu-1 and CP-shu-2.

**Figure 3 nanomaterials-12-04034-f003:**
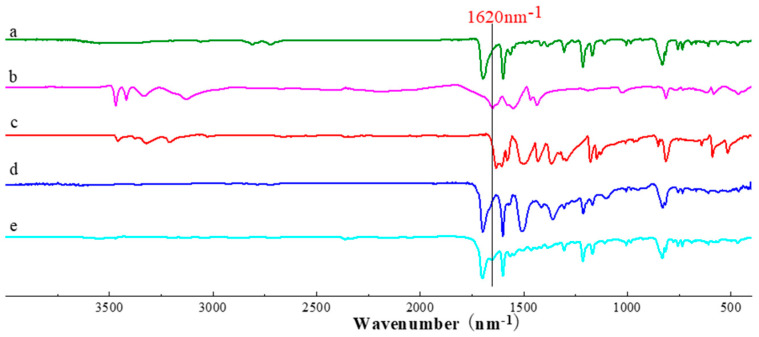
FT-IR spectra of (**a**) TFFPy, (**b**) melamine, (**c**) TTTA, (**d**) CMP-shu-1 and (**e**) CMP-shu-2.

**Figure 4 nanomaterials-12-04034-f004:**
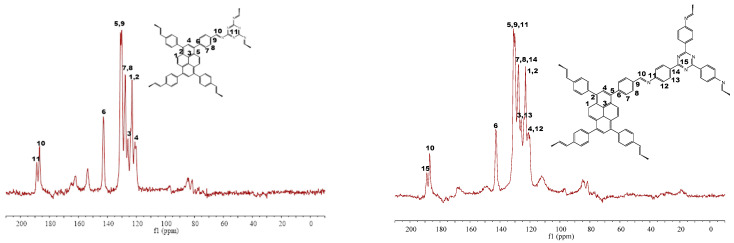
Solid-state ^13^C CP-MAS NMR spectrum of CMP-shu-1 (**left**) and CMP-shu-2 (**right**). The number in the structure figure means the carbon types, to correspond to the CP-MAS.

**Figure 5 nanomaterials-12-04034-f005:**
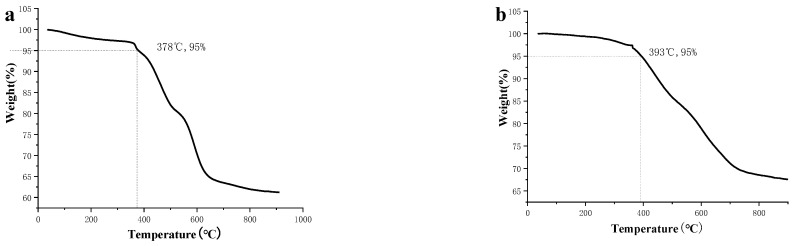
TGA profiles of (**a**) CMP-shu-1, (**b**) CMP-shu-2.

**Figure 6 nanomaterials-12-04034-f006:**
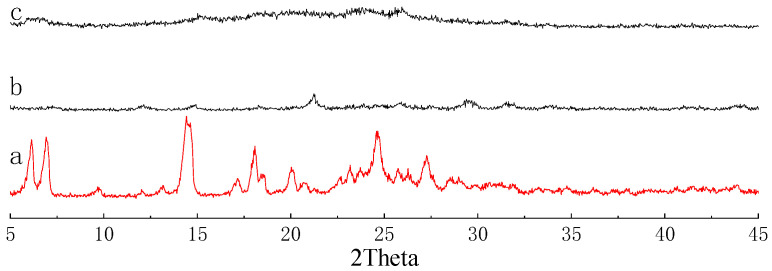
PXRD patterns of (**a**) TFFPy, (**b**) CMP-shu-1, (**c**) CMP-shu-2.

**Figure 7 nanomaterials-12-04034-f007:**
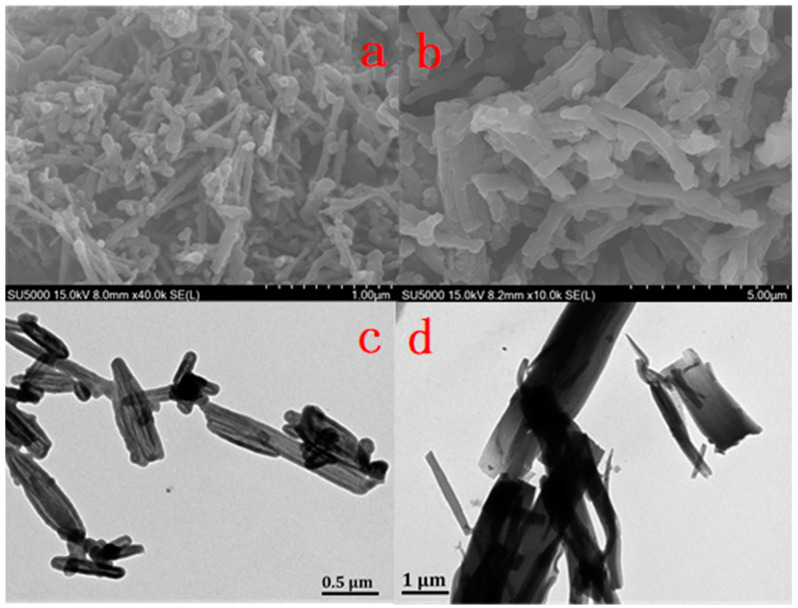
SEM images for (**a**) CP-shu-1 and (**b**) CP-shu-2, respectively. TEM images for (**c**) CPshu-1 and (**d**) CP-shu-2.

**Figure 8 nanomaterials-12-04034-f008:**
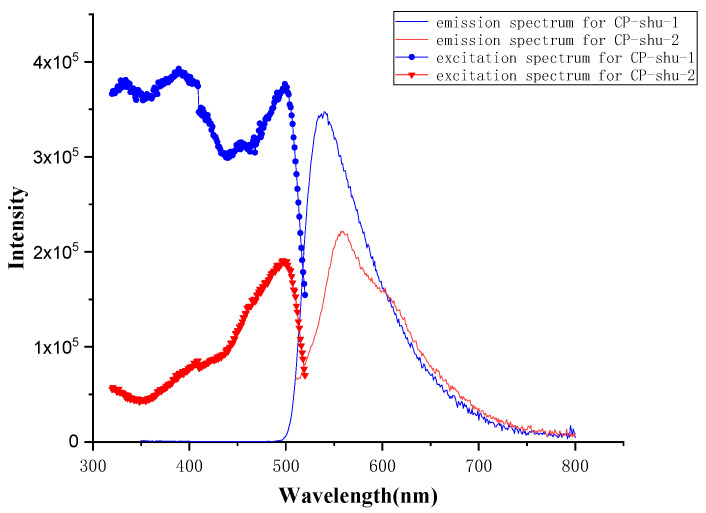
Fluorescence emission and excitation spectra for CP-shu-1 (blue) and CP-shu-2 (red).

**Figure 9 nanomaterials-12-04034-f009:**
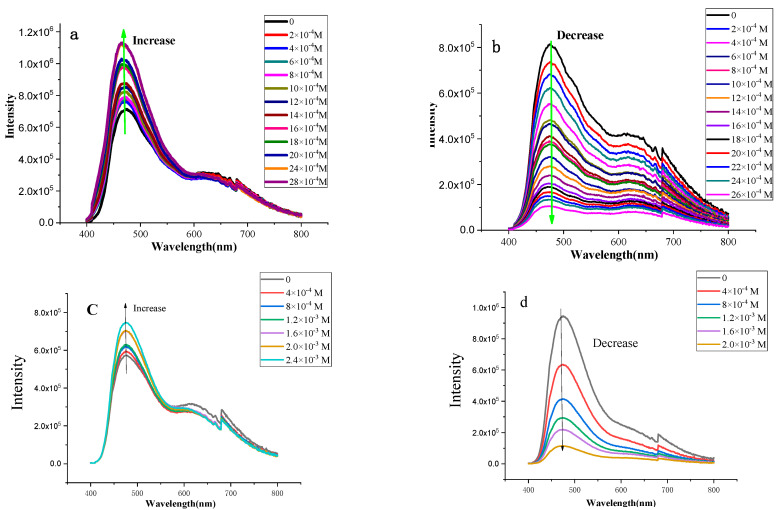
Fluorescence titration for (**a**) benzene, (**b**) nitrobenzene into CP-shu-1 dispersed in acetonitrile, and fluorescence titration for (**c**) benzene, (**d**) nitrobenzene into CP-shu-2 dispersed in acetonitrile.

**Figure 10 nanomaterials-12-04034-f010:**
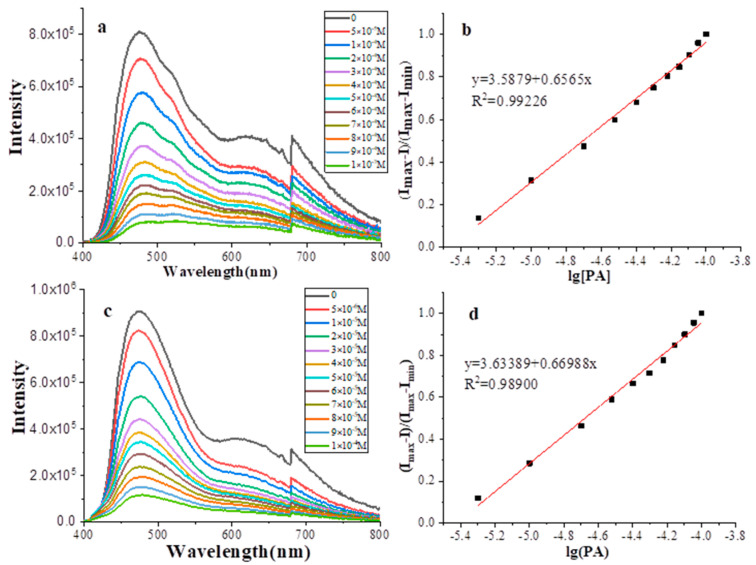
Fluorescence titration spectra of (**a**) CP-shu-1 and (**c**) CP-shu-2 in the presence of different amounts of PA in acetonitrile and the corresponding (I_max_ − I)/(I_max_ − I_min_) vs. lg[PA] plot (**b**,**d**) for the fluorescence titration.

**Figure 11 nanomaterials-12-04034-f011:**
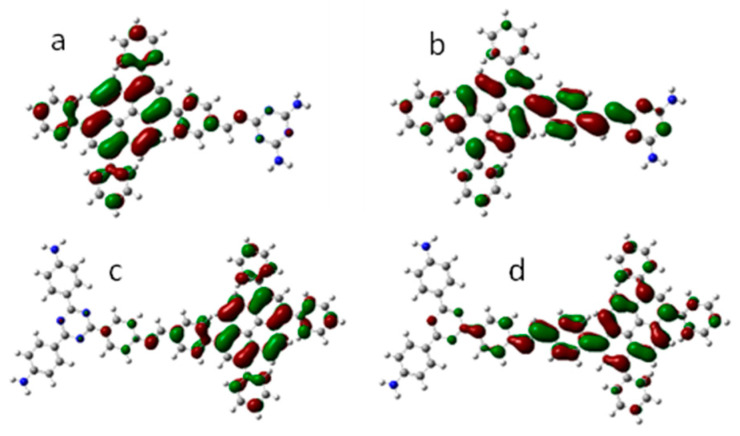
HOMO orbitals for (**a**) CP-shu-1 and (**c**) CP-shu-2 and LUMO orbitals for (**b**) CP-shu-1 and (**d**) CP-shu-2.

**Figure 12 nanomaterials-12-04034-f012:**
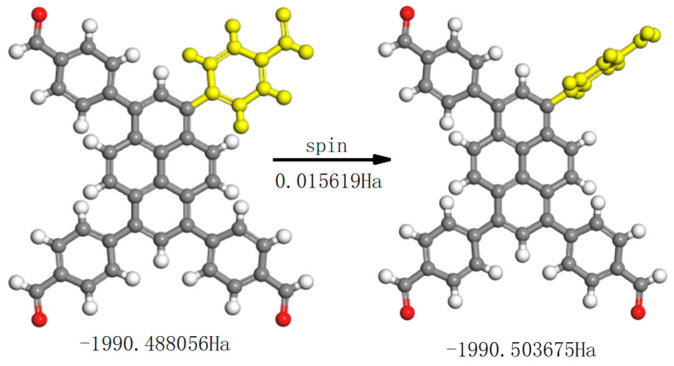
The benzene ring spin energy of TFFPy.

## Data Availability

Not applicable.

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
