# Peer review of "A Novel 3D-Morphology Pyrene-Derived Conjugated Fluorescence Polymer for Picric Acid Detection"

_nanomaterials, 2022, doi:10.3390/nano12224034_

Round 1

Reviewer 1 Report

Review of the manuscript entitled ‘ A novel 3D-morphology pyrene-derived conjugated fluorescence polymer for picric acid detection’

The manuscript is well written even if it contains some grammatical mistakes. However, the results show that the material can be used for sensing picric acid. The study combines experiments and DFT calculations.

It seems more like nanorods than nanofibers (fig. 7). So why do say that ‘…the average rod diameter of CP-shu-1 is about 100nm…’

Author Response

We have changed the description of all "nanofibers" into nanorod. Thanks for your advise. This is our mistake to define the morphology as "nanofiber", it should be nanorod.

Reviewer 2 Report

Please find an attached pdf.

Author Response

We have checked the experiment record and made the yield data into correct. Also the graphic lable has been corrected. The cited references  have been improved. Thank you for your advise.

Reviewer 3 Report

- In the Introduction part, the information about ACQ should be fully elaborated.

- The solubility and/or dispersion properties of the prepared CPs should be examined. Also, the solution UV absorption and solution emission properties are needed to be presented.

- Figure 8 is not enough for ACQ behaviors. The authors should present more evidence for the ACQ behaviors of the CPs.

- The authors commented that "Before titration, the CPs were dispersed in THF via ultrasonic for 10 minutes to make a 0.2 mg/mL suspension". UV and PL spectra of CPs under different concentrations should be presented.

- Binding constant of CP-1 and CP-2 with picric acid should be estimated.

- Rephrase the following (two "further" in a sentence): "The spin energy of TFFPy was further calculated to further investigate the rotating of the benzene arm".

Author Response

We have replyed for all the advices in this document. Thank you for  your review and advices.
